# A new amino acid substitution in the MvALS1 gene of metsulfuron-methyl resistant biotypes Monochoria vaginalis (Burm. f.) C. Presl from West Java, Indonesia

**Kansa Dianti Putri** ⓘ *, **Dwi Guntoro***, **Sintho Wahyuning Ardie**, **Hariyadi**

Department of Agronomy and Horticulture, IPB University, Bogor, Indonesia

* kansadianti@apps.ipb.ac.id (KDP); dwi_guntoro@apps.ipb.ac.id (DG)

## Abstract

The most bothersome weed in rice fields in the Indonesian province of West Java is *Monochoria vaginalis* (Burm. F.) C. Presl, an aquatic herbaceous plant. Metsulfuron-methyl has long been used in wetland rice in West Java with a high enough intensity. However, the case of *Monochoria vaginalis* resistance to metsulfuron-methyl herbicides in Indonesia has not been widely reported and investigated. The study aims to (1) classify the resistance level of *M. vaginalis* toward metsulfuron-methyl, (2) identify Target Site Resistance (TSR) mechanism mutations in the *MvALS1* gene of the resistant biotype of *M. vaginalis*. The Whole Plant Pot Test method was utilized to assess the resistance level of *Monochoria vaginalis*. Following that, all samples were subjected to DNA sequencing using the PCR method to identify mutations in the *MvALS1* gene from the resistant biotype. After then, this study used DUET, a server with an integrated computational methodology, to anticipate the effect of mutations on protein stability. The result showed that *Monochoria vaginalis* from Rawamerta, Karawang showed a moderate level of resistance to metsulfuron-methyl with a resistance ratio of 6.00, Patokbeusi, Subang showed a low level of resistance to metsulfuron-methyl with a resistance ratio of 3.89, compared to susceptible *Monochoria vaginalis*. Nucleotide base alignment in the *MvALS1* gene revealed that base substitutions occurred in the *Monochoria vaginalis* biotype from Rawamerta and Patokbeusi, resulting in 5 amino acid substitutions: Ser-64-Ala, Asp-66-Glu, Asn-240-Asp, Glu-426-Asn, and Ser-469-Asn and Sukra: Ser-64-Ala, Asp-66-Glu, and Asn-240-Asp. The analysis showed that S64A, D66E, and N240D stabilize the protein, whereas E426N and S469N destabilize it. This study confirms for the first time that Ser-64-Ala, Asn-240-Asp, and Glu-426-Asn amino acid mutations were found in cases of *M. vaginalis* resistance to metsulfuron-methyl (ALS inhibitor).

## Introduction

Consumption of West Java domestic rice is the highest in Indonesia 3.85 million tons or about 18.64% of total household consumption [1]. The three districts with the largest rice production

**Data Availability Statement:** All relevant data are within the manuscript and its Supporting Information files.

**Funding:** Kansa Dianti Putri 202207110310741 was granted funding by LPDP - Educational Fund Management Institution to support the published article (https://lpdp.kemenkeu.go.id/). The funding was given to the scholarship recipient who was accepted into the Q1 Journal, with no involvement in the paper itself.

**Competing interests:** The authors have declared that no competing interests exist.

in West Java namely Indramayu Regency 1.42 million tons, Karawang Regency 1.10 million tons and Subang Regency 1.02 million tons [2]. However, some efforts to maintain and increase rice yields often encounter several problems, including competition between rice plants and weeds.

Weed species are the most dominant in rice fields are *Monochoria vaginalis*, *Echinochloa glabrescens*, and *Ludwigia octovalis* [3]. *Monochoria vaginalis* is possible causing rice yield to decrease by up to 82% and *M. vaginalis* in density 150 m$^{-2}$ can reduce rice yield by about 25% in Indonesia [4, 5]. The highest dominance of weeds in rice fields in Subang, West Java is *Monochoria vaginalis* with a total value dominance (NJD) is 35.89% [6]. *Monochoria vaginalis* spreads by vegetative propagation and is a C3 plant, similar to rice [7].

Currently, herbicides are widely used to control weeds in Indonesia. Farmers use herbicides due to a lack of workers willing to work in agricultural land and the difficulty in finding personnel to control weeds mechanically. The chemical use of herbicides is one way to control weeds in rice cultivation. The utilization of herbicides is 80% more profitable than conventional weeding [8]. Metsulfuron-methyl (sulfonylurea group) is a broadleaf weed control herbicide that inhibits enzim acetolactate synthase (ALS). It is widely used in rice fields in Indonesia. ALS (E.C. 2.2.1.6) catalyzes the first step in the biosynthesis of the branched-chain amino acids valine (Val), leucine (Leu), and isoleucine (Ile) [9]. However, the extensive use of herbicides will cause weed to evolve resistance. Weed resistance occurs frequently over a long period due to the utilization of the same herbicide or herbicide with the same mode of action [10]. *M. vaginalis* from Karawang and Subang (Indonesia) are known to exhibit metsulfuron-methyl resistance with R/S of 3.05 and 2.14, meaning the dose used to inhibit 50% of the population exceeds the recommended dose of 4 g a.i. ha$^{-1}$ [6].

The resistance mechanism is based on the level of involvement of the target protein, specifically target site resistance (TSR), which is related to mutations at the target sites of herbicide action and resistance to non-target related (NTSR) location with secondary metabolic pathways [11]. The plant ALS enzyme, encoded by the Als gene, includes at least three copies of *MkALS* in *Monochoria korsakowii* and five copies of *MvALS* in *M. vaginalis* [12]. Point mutations in the gene encoding ALS are thought to cause herbicide resistance. A Japanese population of M. *vaginalis* was found to be resistant to the herbicide, and the changes in the amino acids Pro-197-Ser and Asp-376-Glu were caused by two point mutations in the *MvALS1* gene. *MvALS1* gene expression was reported to be higher than other *MvALS* in *M. vaginalis* [13], thus mutations in *MvALS1* gene are likely to cause herbicide resistance through the TSR mechanism. Weed resistance to herbicides may cause problems later due to lack of knowledge and database, such as increasing herbicide dosage and intensity, which would encourage Weed herbs to thrive and become more resilient species. Therefore, this experiment was conducted to (1) classify the resistance level of *M. vaginalis* toward metsulfuron-methyl, (2) identify Target Site Resistance (TSR) mechanism mutations in the *MvALS1* gene of the resistant biotype of *M. vaginalis*.

## Materials and methods

### Plant materials and growth conditions

Seeds of suspected resistant *M. vaginalis*. populations were collected from rice fields in Patrol (6°18'53.0"S 108°00'19.2"E), Sukra (6°18'41.2"S 107°58'46.2"E), Rawamerta (6°15'12.7"S 107°20'30.3"E), Karawang Timur (6°18'22.0"S 107°19'35.4"E), Patokbeusi (6°20'54.6"S 107°38'51.4"E), and Ciasem (6°18'30.6"S 107°42'56.5"E) districts. *M. vaginalis*. seeds were also obtained from herbicide-untreated paddy fields in Tambakdahan (6°21'39.9"S 107°48'07.8"E), subsequently referred to as susceptible biotype. The location was chosen based on reports

from farmers in the three largest rice producing districts in West Java [2]. Permits or approvals were not required for this work because the samples were collected in public places, with no private or restricted access involved. Furthermore, the sampling excluded any protected species, removing the requirement for specific conservation licenses. The Whole Plant Pot Test system [14] is the method used in this study. The plants were cultivated in plastic pots (20 cm in diameter) that were filled with rice paddy field soil. The experimental design consisted of a split-plot design with two factors and four replications for each experiment. The first factor was the origin of weeds, as the main plot contained seven weed locations, and the second factor was the metsulfuron-methyl 20% dose.

## Dose-response experiment

Seedlings have been thinned to 20 plants pot$^{-1}$ before herbicide treatment. At the 2- to 3-leaf stage, herbicides were applied using a Bengawan Solo 425 Compression Sprayer Manual 15 L, Lurmark flat fan green nozzle 1.2 L min$^{-1}$. Metsulfuron-methyl was applied at 0, 1, 2, 4, 8, 16, and 32 g a.i. ha$^{-1}$. These rates are equivalent to 0, 0.25 0.5, 1, 2, 4, and 8 times of the recommended field rate of the products. The recommended rate of metsulfuron-methyl is 4 g a.i. ha$^{-1}$, respectively. The plants were cut on the soil surface twenty-one days after treatment (DAT), oven-dried for 48 h at 80°C [15], and shoot dry weights were recorded.

## Data analysis

The whole-plant dose-response data were exposed to ANOVA utilizing SPSS v. 25.0. Non-linear regressions of equation (1) were used to analyze data [16]:

$$y = C + \frac{D - C}{1 + (x/I_{50})^b} \tag{1}$$

where: C = lower limit, D = upper limit, b = slope, and $I_{50}$ = a dosage that gives a 50% response

An analysis of the dose-response curves was performed using Origin Pro 2016 [17]. For each herbicide, the Resistance-Susceptible (R/S) ratio was determined as follows: $GR_{50}$ of the resistant (R) population/$GR_{50}$ of the susceptible (S) population. According to [18], herbicide sensitivity is further divided into five classes: sensitive (R/S< 2.0), reduced sensitivity (R/S = 2.0–2.9), low resistance (R/S = 3.0–4.9), moderate resistance (R/S = 5.0–9.9), high resistance (R/S = 10.0–68.1), and very high resistance (R/S> 68.2).

## Isolation of DNA and gene sequence

Genomic DNA was isolated from the leaves of susceptible and resistant biotypes of *M. vaginalis*. Leaves weighing 0.1–0.5 g were mashed together with 700 L of DNA extraction buffer solution [100 mM Tris HCl (pH 8), 1.4 M NaCl, 20 mM EDTA, 2% CTAB, and 1% PVP] [19]. The DNA quality was determined using 1% agarose, and the quantity of DNA was tested using a Thermo ScientificTM DropTM and Multiskan SkyHigh Microplate Spectrophotometer, with the results shown in S1 Table. A primer set 5'-ATGGCTGCTTCGAAGCCCTCTCCATT-3' (forward) and 5' ACTAGTGCACTGTGCTCCCATCTCCAT-3' (reverse) was used to amplify DNA with a length of 1.931 bp at the position between 1 base and 1.931 base in the gene encoding *MvALS1* (GenBank accession number AB243613.1). Amplification was carried out by PCR in a total volume of 50 μL containing 25 μL 2x KAPA2G Fast ReadyMix Kit, 2.5 μL forward and reverse primers with a concentration of 10 μM each, 12.5 μL of *M.vaginalis* genomic DNA with a concentration of 2 ng μL$^{-1}$ and 7.5 μL Nuclease Free Water (NFW). The PCR conditions were as follows: denaturation at 94°C for 2 min; 40 cycles of 94°C for 5 s, 60°C for 30 s;

elongation temperature at 72˚C for 1 min, and final extension at 72˚C for 10 min. The amplification results were analyzed by electrophoresis of 5 μL of PCR product on a 1.5% (w/v) agarose gel at 90 V for 85 min in 1x TAE buffer (Tris-acetate-EDTA). The gel resulting from electrophoresis was then stained with ethidium bromide (0.5 μg mL$^{-1}$) for 10 seconds and continued with visualization using a UV transilluminator (Gel Doc EZTM, Bio-Rad, United States). Amplification of the *MvALS1* gene produced an amplicon of approximately 1.931 bp in each of the seven *M.vaginalis* biotypes in S1 Fig. Nucleotide base tracing was carried out using the Sanger method by the service company PT Genetica Science Indonesia. Alignment of nucleotide base sequences was carried out using the MUSCLE algorithm in Bioedit 7.2.

### Forecasting mutation impact on protein stability

The wild-type protein sequence (FASTA) format was retrieved from the NCBI GenBank under the entry number BAE53594.1, which corresponds to the DBSOURCE accession AB243613.1. To convert this protein sequence into a PDB structure, the SWISS-MODEL webserver was used [20]. In this study, the DUET tool was utilized to examine monomeric structures with a specific point mutation in chain A [21]. DUET combines predictions from two computational approaches, mCSM and SDM, using SVM regression with a Radial Basis Function (RBF) kernel to produce a consensus prediction of the mutation's impact [22]. Furthermore, we employed the COFACTOR web server as a unified platform for structure-based multiple-level protein function predictions, obtaining data such as enzyme commission and predicted active site residues from homologous function templates [23].

## Results

### Sensitivity to ALS herbicides

The dosage response curve (Fig 1) was used to obtain the GR$_{50}$ value for each treatment of population origin and herbicide level. The susceptible biotype's GR$_{50}$ value was 0.51 g a.i. ha$^{-1}$. Meanwhile, two of the six resistant *Monochoria vaginalis* populations, Rawamerta and Patokbeusi, demonstrated resistance to the herbicide metsulfuron-methyl, with a GR$_{50}$ value of 3.09 g a.i. ha$^{-1}$ (dosage 0.77 times higher than the recommended dose) for Rawamerta and GR$_{50}$ 2.00 g a.i. ha$^{-1}$ (dosage 0.50 times higher than recommended dose) for Patokbeusi. The survival percentage for Rawamerta and Patokbeusi biotypes was thought to be higher than the required dose of metsulfuron-methyl, which is >4 g a.i. ha$^{-1}$ to kill 50% of the *Monochoria vaginalis* population.

### Confirmation of resistance

The comparison value between GR$_{50}$ *Monochoria vaginalis* exposed to herbicides and GR$_{50}$ *Monochoria vaginalis* not exposed to herbicides (susceptible) was used to calculate a resistance ratio, and the resistance ratio value is used to assess weed resistance to herbicides [18]. Table 1 shows the resistance level of *Monochoria vaginalis* accessions to the herbicide metsulfuron-methyl. Patrol, East Karawang, and Ciasem biotypes were sensitive to metsulfuron-methyl, with R/S values of 1.59, 1.43, and 1.36, respectively. Meanwhile, *Monochoria vaginalis* from Rawamerta exhibited moderate resistance to metsulfuron methyl (R/S 6.00), indicating that it was 6.00 times more resistant than susceptible *Monochoria vaginalis*. *Monochoria vaginalis* from Patokbeusi showed low resistance to metsulfuron methyl (R/S 3.89), but *Monochoria vaginalis* from Sukra demonstrated low sensitivity (R/S 2.77). This low resistance occurred because it was discovered that raising the dose may still produce a decrease in the average dry weight of *Monochoria vaginalis* from East Karawang. Thus far, 14.29% of the *Monochoria*

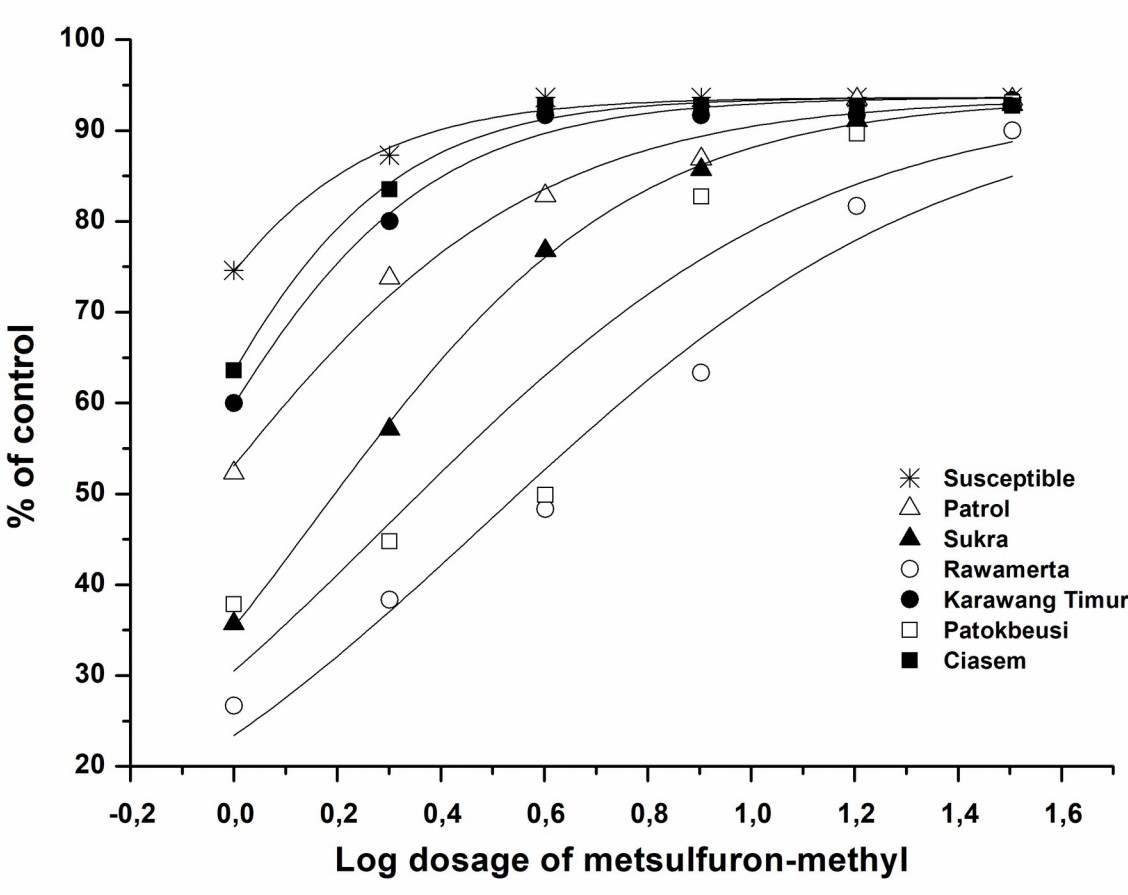

**Fig 1. Inhibition curves by metsulfuron-methyl from seven populations of *M. vaginalis*.** Lines are the reaction bends foreseen by non-linear regression; symbols represent shares of average survival, upheld the untreated controls.

*vaginalis* population from West Java were moderate resistance (R/S = 5.0–9.9), 14.29%, low resistance (R/S = 3.0–4.9), 14.29% decreased sensitivity (R/S = 2–2.9) and 57.14% sensitive (R/S = <2) to metsulfuron-methyl.

**Table 1. Metsulfuron-methyl dose needed to kill 50% of plants (GR$_{50}$) and R/S ratios obtained from the curves of population S and R *M. vaginalis*.**

| Biotype | GR$_{50}$ (g b.a ha$^{-1}$) | *Adj. R-square* | R/S | Resistance Category [18] |
|---|---|---|---|---|
| Suscpetible | 0.51 | 0.98 | - | - |
| Patrol | 0.82 | 0.98 | 1.59 | Susceptible |
| Sukra | 1.42 | 0.99 | 2.77 | Reduced Sensitivity |
| Rawamerta | 3.09 | 0.95 | 6.00 | Moderate Resistance |
| Karawang Timur | 0.73 | 0.98 | 1.43 | Susceptible |
| Patokbeusi | 2.00 | 0.86 | 3.89 | Low Resistance |
| Ciasem | 0.70 | 0.99 | 1.36 | Susceptible |

[a]GR$_{50}$ refers to the measurements of herbicide success in retaining 50% dry weight and is expressed as active ingredient grams per hectare (g a.i. ha$^{-1}$). The results of the nonlinear regression analysis of the log-logistic model in S2 Table.

[b]R/S, index Resistance was computed as the ratio of GR$_{50}$ R to GR$_{50}$ S.

The *Monochoria vaginalis* weed population from Rawamerta is classified as moderately resistant with a resistance ratio value of 6.00, Patokbeusi as low resistance with a resistance ratio value of 3.89, and Sukra as decreasing sensitivity with a resistance ratio value of 2.77 to the herbicide metsulfuron-methyl (Table 1).

## Identification of the mutation in *MvALS1* genes

The nucleotide sequences of the 1.931 bp region from the resistant *M. vaginalis* biotype differed by a single nucleotide substitution from the susceptible biotype. The alignment of the *MvALS1* gene sequence in the seven *M. vaginalis* genotypes reveals changes in nucleotide bases for the third genotype (Fig 2). First, *M. vaginalis* from Rawamerta was classified as moderately resistant to metsulfuron-methyl due to changes in nucleotide bases at 8 positions: 190, 198, 381, 681, 718, 1,276, 1,278, and 1,406. Second, *M. vaginalis* from Patokbeusi had low resistance and experienced changes in nucleotide bases at six sites: 190, 198, 718, 1,276, 1,278, and 1406. Finally, *M. vaginalis* from Sukra was classified as having decreased sensitivity and had nucleotide base changes at four sites: 190, 198, and 199 (Fig 2). In contrast, no nucleotide differences between sensitive *M. vaginalis* were found in *MvALS1*. The gene sequence revealed that the susceptible and resistant biotypes differed by a single nucleotide.

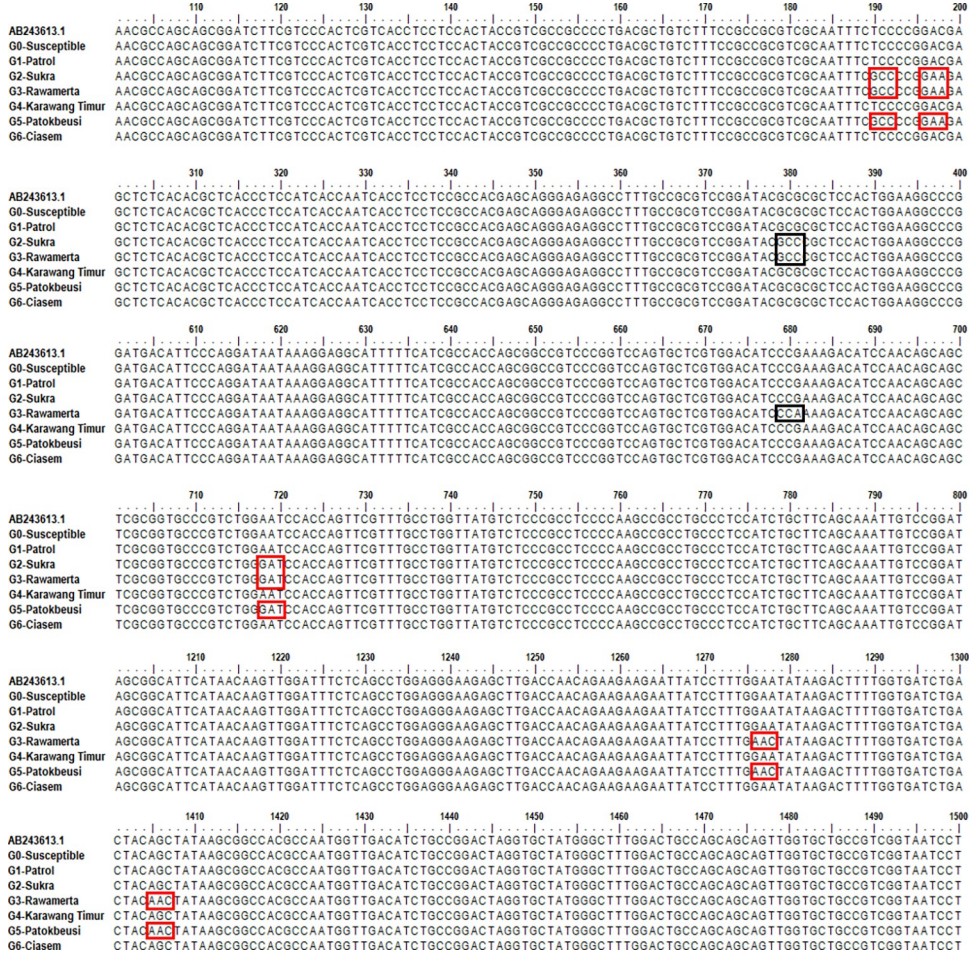

**Fig 2. A single base mutation in the full sequence of the *MvALS1* gene.** The red box represents nucleotide substitutions and the resulting amino acid changes.

The *MvALS1* gene sequences of seven *M. vaginalis* biotypes were compared to those of *M. vaginalis* (GenBank accession no. AB243613.1). Based on the alignment results, those nucleotide base substitutions produced 5 amino acid changes in the weed *Monochoria vaginalis* from Rawamerta and Patokbeusi that were resistant to the herbicide metsulfuron-methyl (ALS inhibitor), at positions 190 (Ser-64-Ala, TCC; sensitive, GCC; resistant), 198 (Asp-66-Glu, GAC; sensitive, GAA; resistant), 718 (Asn-240-Asp, AAT; sensitive, GAT; resistant), 1.276–1.278 (Glu-426-Asn, GAA; sensitive, AAC; resistant), and 1.406 (Ser-469-Asn. AGC; sensitive, AAC; resistant) (Fig 3). *Monochoria vaginalis* from Sukra was then subjected to three single amino acid substitutions at the same positions: 90 (Ser-64-Ala, TCC; sensitive, GCC; resistant), 198 (Asp-66-Glu, GAC; sensitive, GAA; resistant), and 718 (Asn-240-Asp, AAT; sensitive, GAT; resistant). Meanwhile, nucleotide base substitutions at positions 381 (GCG; sensitive, GCC; resistant) and 681 (CCG; sensitive, CCA; resistant) did not result in amino acid changes (Fig 3). Furthermore, the results of nucleotide base substitutions in the *MvALS1* gene showed that the *Monochoria vaginalis* biotype from Rawamerta and Patokbeusi had 5 amino acid substitutions: Ser-64-Ala, Asp-66-Glu, Asn-240-Asp, Glu-426-Asn and Ser-469-Asn and *Monochoria vaginalis* from Sukra experienced 3 single amino acid substitutions: Ser-64-Ala, Asp-66-Glu and Asn-240-Asp (Fig 3). The signal quality chromatogram detected from genotype amplicons that undergo nucleotide base substitution can be seen in S2 Fig. It implies that the above-mentioned amino acid substitution is to blame for SU-herbicide resistance in the R biotype.

## Protein stability

Table 2 shows the number of destabilizing and stabilizing mutations, organized by mutation type. Stabilizing mutations have a positive ΔΔG (kcal mol$^{-1}$), whereas destabilizing mutations have a negative ΔΔG. According to DUET's expected stability alterations, the mutations S64A, D66E, and N240D primarily produce stabilizing mutations (0.283 kcal mol$^{-1}$, 0.086 kcal mol$^{-1}$, and 0.038 kcal mol$^{-1}$). In contrast, the E426N and S469N mutations are largely destabilizing (-0.439 kcal mol$^{-1}$ and -0.402 kcal mol$^{-1}$).

## Discussion

Metsulfuron-methyl herbicide, works by transporting it from the xylem and phloem to all weed tissues and acting as an inhibitor of the acetolactate synthase enzyme, which blocks the

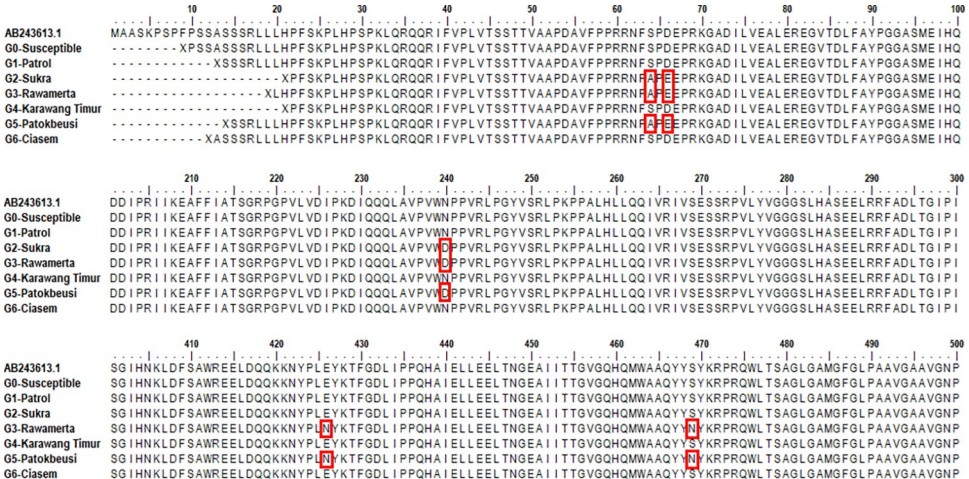

**Fig 3. Alignment of *MvALS1* gene deduced amino acid sequences.** The red box depicts amino acid changes in comparison to the susceptible biotype (Ser-64-Ala, Asp-66-Glu, Asn-240-Asp, Glu-426-Asn, and Ser-469-Asn).

**Table 2. Performance of protein stability prediction method.**

| Mutation | Predicted Stability Change ΔΔG (kcal mol$^{-1}$) | | |
|---|---|---|---|
| | mCSM | SDM | DUET |
| S64A | -0.222 | 1.13 | 0.283 |
| D66E | -0.212 | -0.13 | 0.086 |
| N240D | 0.114 | -0.21 | 0.386 |
| E426N | -0.585 | -0.48 | -0.439 |
| S469N | -0.645 | -0.16 | -0.402 |

[a]Mutations: + ΔΔG (kcal mol$^{-1}$), Destabilizing mutations:—ΔΔG (kcal mol$^{-1}$).

[b]SDM (Site-Directed Mutator).

[c]mCSM (mutation Cutoff Scanning Matrix).

[d]Pearson Correlation Coefficient (R): mCSM vs SDM = 0.2342801 (relatively weak); mCSM vs DUET = 0.9497737 (very strong); SDM vs DUET = 0.5219668 (moderate).

production of the three amino acids valine, leucine, and isoleucine, thereby inhibiting plant shoot and root cell division [24]. Metabolism is disrupted, resulting in poisoning and even death. In some resistant weeds, inhibiting ALS enzyme formation disrupts protein formation and photosynthetic transmission, resulting in lower biomass and seed yield [25].

*Monochoria vaginalis* weeds from herbicide-resistant rice fields are more difficult to control than weeds from herbicide-free rice fields (Table 1). The percentage of growth damage describes how much the metsulfuron-methyl herbicide dose level can reduce *M.vaginalis* growth from each area of origin (Fig 1). The level of phytotoxicity, percentage of necrotic tissue, and weed death caused by herbicides were used to assess resistance levels. As a result, minor differences such as reduced size and leaf area, to name a few, are not visible. GR$_{50}$ data is a more concrete evaluation, particularly in dose-response curves and when weeds have low levels of resistance [26]. The level of resistance must be determined because it provides an overview of the resistance mechanisms that occur, allowing appropriate strategies to be developed to control resistant weeds [14].

Based on the results of previous research in Indonesia, the GR50 value of *E. crus-galli* from East Karawang was 7.59 g a.i. ha$^{-1}$ metsulfuron-methyl. The GR50 value of metsulfuron-methyl was 1.90 times higher than the recommended application dose (4 g a.i. ha$^{-1}$) [27], and the resistance factor against *Chamaesyce maculate* from Georgia was 90 times higher than susceptible biotypes [28]. Then, in Subang, *M. vaginalis* exhibited a GR50 value of 51.93 g a.i. ha$^{-1}$ bensulfuron-methyl (sulfonylurea), which was 13 times higher than the recommended dose (4 g a.i. ha$^{-1}$) [15]. In addition, *M. vaginalis* had a GR50 value of 9.50 g a.i. ha$^{-1}$ bensulfuron-methyl (sulfonylurea) in Korea, which was 2.38 times higher than the recommended dose (4 g a.i. ha$^{-1}$) [29].

Weed resistance will continue to rise as monoculture practices and herbicides with the same mode of action are used [30]. Long-term herbicide use does not eliminate all weeds, and a small number of weeds will survive and evolve into resistant weed biotypes [31]. Additionally, natural selection a process in which genetic mutations occur naturally in plants due to the continuous use of similar herbicides increasing weed resistance [32]. Differential herbicide sensitivity could be explained by differences in absorption, translocation, or metabolism rates [33].

In ALS, there is a regulatory subunit and a catalytic subunit. By producing amino acids such as valine, leucine, and isoleucine, regulatory subunits regulate catalytic subunit activity, including feedback inhibition. The herbicide cannot mimic the substrate for the enzyme,

according to an analysis of the three-dimensional structures of ALS, the substrate, and the herbicide molecule [15, 34]. ALS inhibitor herbicides thus bind across the catalytic site of the catalytic subunit, preventing substrate access to the ALS catalytic entry point [34]. These herbicides may bind differently to the ALS protein's catalytic sites, resulting in different efficiencies for preventing these herbicide substrates from accessing the catalytic subunit. Furthermore, SU and TP herbicides have been shown to bind to different ALS protein locations [34].

Herbicide resistance mechanisms are classified into two types based on the level of involvement of the target protein: target-site resistance (TSR) mechanisms and non-target-site resistance (NTSR) mechanisms. NTSR mechanisms include reduced herbicide uptake and translocation as well as increased herbicide metabolism to break down into less toxic compounds [35]. The issue of Non-Target Site Resistance (NTSR) in *M. vaginalis* is probably overlooked. Molecular understanding of NTSR processes is still in its early stages [36]. Cytochrome P-450 enzymes, such as those found in *G. coronaria* and *S. trifolia* [37, 38], as well as glutathione S-transferases (GSTs) found in *S. juncoides* [39], play critical roles in herbicide metabolism in ALS (SU) resistant broad-leaf weeds. The CYP81 family contains the majority of CYP450 resistance-related sequences, while resistance sequences within GSTs are primarily found in the GSTU18 and GSTF6 families [40]. However, the potential genes for *M. vaginalis* have not yet been examined in this present research, which focuses solely on the ALS TSR mechanism.

TSR mechanisms, on the other hand, alter the amino acid sequence and expression level of the target enzyme, reducing the herbicide's ability to inhibit the enzyme or necessitating higher herbicide concentrations to achieve adequate inhibition [35]. In most cases of ALS-resistant weeds, single amino acid substitutions (TSR) have been reported. Previous research demonstrated that mutations in Pro-197 result in SU resistance [41]. Resistant weeds emerged only a few years after the introduction of sulfonylurea group herbicides (ALS inhibitors) to the herbicide market. Since then, resistant weeds have emerged, Ala122, Pro197, Trp574, or Ser653 are the most common amino acid substitutions caused by single point mutations [34].

The amino acid mutations in this study, Ser-64-Ala, Asp-66-Glu, Asn-240-Asp, Glu-426-Asn, and Ser-469-Asn (Fig 3), have the potential to increase global understanding of the process and evolutionary patterns of resistance spread and help consider weed management to suppress the potential spread and evolution of resistant weeds. The findings of this study are consistent with previous research, but at different nucleotide base positions for cases of mutations in weeds resistant to ALS enzyme inhibitor SU herbicides (bensulfuron-methyl), where the Asp-376-Glu *ALS1* amino acid substitution previously occurred in *Monochoria vaginalis* [42]. Furthermore, Ser-653-Asn has been discovered in the weed *Amaranthus tuberculatus* (resistant to imazethapyr, pyrithiobac, and thifensulfuron) which were resistant to ALS inhibitor herbicides [43]. Previous research in Indonesia reported the amino acid substitutions Val-143-Ile and Val-148-Ile in the *ALS1* protein with a partial sequence for the first time [15]. The ALS-resistant biotype in *M. vaginalis* then experienced a single mutation in the *ALS1* gene, changing position 197 proline (CCT) to serine (TCT) in Japan [29, 42], position 376 asparagine (GAT) to glutamine (GAA) [13], and Trp-574-Leu [41].

This study used Site DUET to assess the effects of mutations (Ser-64-Ala, Asp-66-Glu, Asn-240-Asp, Glu-426-Asn, Ser-469-Asn) on the stability of the ALS structure. Three mutations (Ser-64-Ala, Asp-66-Glu, Asn-240-Asp) stabilized the protein with $\Delta\Delta G$ values 0.283 kcal mol$^{-1}$, 0.086 kcal mol$^{-1}$, and 0.038 kcal mol$^{-1}$, while Glu-426-Asn and Ser-469-Asn destabilized it with $\Delta\Delta G$ values of -0.439 and -0.402 kcal mol$^{-1}$, respectively (Table 2). Stabilizing mutations keep proteins functional and improve weed resistance, while destabilizing mutations can impair protein structure or interfere with herbicide binding site [44].

However, the findings of this study are consistent with those of prior research [45]. The mutation is unlikely to have an adverse effect on protein function. All of the proposed mutations result in changes outside the active site (S3 Table). Then, this research used COFACTOR with the GenBank accession BAE53594.1 (DBSOURCE accession AB243613.1), the top enzyme homolog identified was PDB Hit 1ybhA, with a CscoreEC of 0.680, a TM-score of 0.902, an RMSDa of 0.75, an IDENa of 0.746, a coverage of 0.906, an EC Number of 2.2.1.6, and active site residues at positions H615, V457, G483, and M485 (S3 Table). Furthermore, based on earlier research, mutations in *Arabidopsis thaliana* ALS's active site channel, such as M124, P197, R199, M200, K256, D257, Q260, D376, K381, E383, W574, F578, Y579, P652, S653, and G655, impact ALS inhibitor binding [34]. W574 directs the active-site channel and anchors herbicides. The aromatic sulfonylureas ring comes into contact with P197 at the active-site channel entry [34]. These mutations affect binding affinity and effectiveness, rendering ALS-inhibiting herbicides ineffective.

Nevertheless, herbicides are often absorbed by weeds before they reach the target of the active site [46]. The degree of this metabolism influences weed control. Herbicide resistance can be increased by genetic modifications in metabolism [46]. Other studies discovered resistance to tribenuron-methyl in *S. alba*, indicating that the herbicide is unable to interact with the target site (ALS) [47].

This study was successful in identifying three previously unknown amino acid substitutions in the *MvALS1* gene, namely Ser-64-Ala, Asn-240-Asp, and Glu-426-Asn in the *M. vaginalis*. These findings suggest that resistance to metsulfuron-methyl in the three *M. vaginalis* biotypes studied (Rawameta, Patokbeusi, and Sukra) was caused by the target site resistance (TSR) mechanism, specifically changes in nucleotide bases in the *MvALS1* gene and metsulfuron-methyl was no longer effective in those three areas. Gaines et al. [35] discovered that non-synonymous substitutions that alter amino acid translation can result in target site resistance. Further research into resistance mechanisms from different regions leads to a more complete understanding of the evolution of the *MvALS1* gene, which causes resistance in *M. vaginalis*.

## Conclusion

*Monochoria vaginalis* populations from Rawamerta were classified as moderately resistant with a resistance ratio value of 6.00, Patokbeusi as low resistance with a resistance ratio value of 3.89, and Sukra as decreased sensitivity with a resistance ratio value of 2.77 to the herbicide metsulfuron-methyl, whereas *M. vaginalis* originating from the from Patrol, East Karawang and Ciasem are classified as sensitive weeds with resistance ratios of 1.59, 1.43 and 1.36 to metsulfuron methyl, respectively. Alignment of the nucleotide bases in the *ALS1* gene reveals that the *Monochoria vaginalis* biotypes from Rawamerta and Patokbeusi experienced nucleotide substitution, resulting in 5 amino acid substitutions: Ser-64-Ala, Asp-66-Glu, Asn-240-Asp, Glu-426-Asn, and Ser-469-Asn. Then, *Monochoria vaginalis* from Sukra underwent nucleotide substitution, resulting in three amino acid substitutions: Ser-64-Ala, Asp-66-Glu, and Asn-240-Asp. The analysis of protein stability owing to various mutations revealed that S64A, D66E, and N240D mutations stabilized the protein structure, instead E426N and S469N mutations destabilized it. In contrast, *M. vaginalis* from previous bioassays that were still classified as sensitive (Patrol, East Karawang, and Ciasem) did not show any changes in nucleotide bases. This is the first reported case of *M. vaginalis* developing resistance to metsulfuron-methyl (sulfonylurea). This also confirms for the first time that Ser-64-Ala, Asn-240-Asp, and Glu-426-Asn amino acid mutations were found in cases of *M. vaginalis* resistance to metsulfuron-methyl (ALS inhibitor).

## Supporting information

**S1 Fig. Visualization of the DNA quality obtained from PCR of seven different biotypes of** *Monochoria vaginalis.*
(TIF)

**S2 Fig. Chromatogram of** *MvALS1* **gene readings from Rawamerta, Patokbeusi, and Sukra** *M. vaginalis* **genotypes.**
(TIF)

**S1 Table. Purification of** *M. vaginalis* **results of a DNA spectrophotometer.**
(PDF)

**S2 Table. Origin Pro data result and analysis on metsulfuron-methyl GR$_{50}$.**
(PDF)

**S3 Table. Estimated Enzyme Commission (EC) numbers and active site residues.**
(PDF)

## Author Contributions

**Conceptualization:** Kansa Dianti Putri, Dwi Guntoro, Sintho Wahyuning Ardie.

**Data curation:** Kansa Dianti Putri.

**Formal analysis:** Kansa Dianti Putri.

**Funding acquisition:** Kansa Dianti Putri.

**Investigation:** Kansa Dianti Putri.

**Methodology:** Kansa Dianti Putri, Dwi Guntoro, Sintho Wahyuning Ardie.

**Project administration:** Kansa Dianti Putri.

**Resources:** Kansa Dianti Putri.

**Software:** Kansa Dianti Putri.

**Supervision:** Dwi Guntoro, Hariyadi.

**Validation:** Dwi Guntoro, Sintho Wahyuning Ardie, Hariyadi.

**Visualization:** Kansa Dianti Putri, Dwi Guntoro, Sintho Wahyuning Ardie.

**Writing – original draft:** Kansa Dianti Putri.

**Writing – review & editing:** Kansa Dianti Putri, Dwi Guntoro, Sintho Wahyuning Ardie.

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
