## [Decision Letter · Decision Letter 0]

28 May 2024

PONE-D-24-02160A New Amino Acid Substitution in the MvALS1 Gene of Metsulfuron-Methyl Resistant Biotypes Monochoria vaginalis (Burm. f.) C. Presl From West Java, IndonesiaPLOS ONE

Dear Dr. Putri,

Thank you for submitting your manuscript to PLOS ONE. After careful consideration, we feel that it has merit but does not fully meet PLOS ONE’s publication criteria as it currently stands. Therefore, we invite you to submit a revised version of the manuscript that addresses the points raised during the review process.

We look forward to receiving your revised manuscript.

Kind regards,

R. M. Sundaram, Ph.D.

Academic Editor

PLOS ONE

Journal Requirements:

 [Kansa Dianti Putri 202207110310741 was granted funding by LPDP - Educational Fund Management Institution to support the published article (https://lpdp.kemenkeu.go.id/). The funding was given to the scholarship recipient who was accepted into the Q1 Journal, with no involvement in the paper itself.].  

[We thank LPDP - Educational Fund Management Institution for granting the scholarship and supporting the project.]

 [Kansa Dianti Putri 202207110310741 was granted funding by LPDP - Educational Fund Management Institution to support the published article (https://lpdp.kemenkeu.go.id/). The funding was given to the scholarship recipient who was accepted into the Q1 Journal, with no involvement in the paper itself.]. 

6. We note that Figure 1 in your submission contain copyrighted images. All PLOS content is published under the Creative Commons Attribution License (CC BY 4.0), which means that the manuscript, images, and Supporting Information files will be freely available online, and any third party is permitted to access, download, copy, distribute, and use these materials in any way, even commercially, with proper attribution. For more information, see our copyright guidelines: http://journals.plos.org/plosone/s/licenses-and-copyright.

7. We note that Figure S1 in your submission contain [map/satellite] images which may be copyrighted. All PLOS content is published under the Creative Commons Attribution License (CC BY 4.0), which means that the manuscript, images, and Supporting Information files will be freely available online, and any third party is permitted to access, download, copy, distribute, and use these materials in any way, even commercially, with proper attribution. For these reasons, we cannot publish previously copyrighted maps or satellite images created using proprietary data, such as Google software (Google Maps, Street View, and Earth). For more information, see our copyright guidelines: http://journals.plos.org/plosone/s/licenses-and-copyright.

a. You may seek permission from the original copyright holder of Figure S1 to publish the content specifically under the CC BY 4.0 license.  

Additional Editor Comments:

In view of the comments from both the reviewers, I understand that the manuscript needs a significant improvement through a major revision based on the points raised by both the reviewers. I therefore recommend the manuscript for major revision.

Reviewers' comments:

Reviewer's Responses to Questions

**Comments to the Author**

1. Is the manuscript technically sound, and do the data support the conclusions?

Reviewer #1: Yes

Reviewer #2: Partly

2. Has the statistical analysis been performed appropriately and rigorously? 

Reviewer #1: Yes

Reviewer #2: Yes

3. Have the authors made all data underlying the findings in their manuscript fully available?

Reviewer #1: Yes

Reviewer #2: No

4. Is the manuscript presented in an intelligible fashion and written in standard English?

Reviewer #1: Yes

Reviewer #2: Yes

5. Review Comments to the Author

Reviewer #1: Authors have characterized the herbicide resistance in Monochoria in Indonesia. The introduction and discussion section needs major improvement. The comments are inserted in the attached PDF document. Authors should cut down the management part from discussion section.

Authors should discuss about NTSR role in ALS resistance evolved in Monochoria. In Results section, authors should discuss the findings as depicted in Tables and Figures. The support of peers/researchers can be discussed in the discussion section.

Reviewer #2: The research already found some mutation pattern in the weed that actually conform to the previous research. Also, it elicited ‘Resistance-susceptible” (R/S) ratio to analyze the resistance of the weed toward the inhibitor.

However, there are concerns that should be addressed pertaining this research.

The elicited mutational patterns did not lead us to the conclusion whether they are hampering both the binding and catalytic site of the protein or not. If there are no informations on what happen on those sites, basically we cannot infer anything at all.

There are basically two ways to do this. First: Using protein stability predictors. If the software can predict that the protein is unstable, it means that there is a tendency if the inhibition is not perfect. You can check this paper for more insight:

https://bmcbioinformatics.biomedcentral.com/articles/10.1186/s12859-021-04238-w

Performance of Web tools for predicting changes in protein stability caused by mutations | BMC Bioinformatics | Full Text (biomedcentral.com)

Second: Using molecular docking simulation. Making a comparison between docking of wild-type protein-inhibitor, and mutated-protein-inhibitor. If the free binding energy of the mutated protein is inclined to a more positive value, it means that the inhibition process is weakened. You can check this paper for more insight:

https://pubmed.ncbi.nlm.nih.gov/34914115/

Omicron and Delta variant of SARS-CoV-2: A comparative computational study of spike protein - PubMed (nih.gov)

Based on your current data, you can find the wild type protein in RCSB database, and then use homology modeling to construct the preidcted mutated protein. Then, you can do one of the aforementioned approach to proceed.

6. PLOS authors have the option to publish the peer review history of their article (what does this mean?). If published, this will include your full peer review and any attached files.

Reviewer #1: **Yes: **Simerjeet Kaur

Reviewer #2: No

---

## [Author Response · Author response to Decision Letter 0]

1 Jul 2024

Dear Dr. Sundaram, 

I hope this message finds you well. Thank you for your detailed feedback on our manuscript, PONE-D-24-02160, titled "A New Amino Acid Substitution in the MvALS1 Gene of Metsulfuron-Methyl Resistant Biotypes Monochoria vaginalis (Burm. f.) C. Presl From West Java, Indonesia." We appreciate the opportunity to revise and resubmit our work. Below, we address each point raised by the academic editor and reviewers.

Journal Requirements:

1. Manuscript Style Requirements:

 We have ensured our revised manuscript meets PLOS ONE’s style requirements, including proper file naming. We have used the provided templates:

PLOS ONE Formatting Sample Main Body

PLOS ONE Formatting Sample Title Authors Affiliations

2. Permits Information in Methods Section:

 We have included information in the Methods section regarding the fieldwork, specifying that no permits or approvals were necessary since the samples were gathered in public areas without any private or restricted access. Additionally, our sampling did not involve protected species, eliminating the need for specific conservation permits.

3. Financial Disclosure Statement:

 We kindly request to update the financial disclosure statement as outlined below: This research was supported by LPDP - Indonesia Endowment Fund for Education. The funders had no role in study design, data collection and analysis, decision to publish, or preparation of the manuscript.

4. Acknowledgments Section:

 We have removed any funding-related text from the Acknowledgments section. 

5. Original Uncropped and Unadjusted Images:

 We have ensured compliance with PLOS ONE’s requirements for blot and gel images by providing the original uncropped and unadjusted images. We have indicated in our cover letter whether these images are included in the Supporting Information.

6. Copyrighted Images in Figure 1:

 We have taken the action to remove the figure 1.

7. Copyrighted Images in Figure S1:

 For Figure S1, we have proceeded with the action of removing the figure from the manuscript.

8. Uploading Figure Files to PACE for Compliance Verification

 We have already uploaded our figure files to the Preflight Analysis and Conversion Engine (PACE) digital diagnostic tool and are currently reviewing them. We will ensure that the figures meet PLOS requirements and will include them in the revised submission accordingly.

Review Comments to the Author:

Dear Dr. Simerjeet Kaur,

Thank you for your valuable feedback on our manuscript titled "A New Amino Acid Substitution in the MvALS1 Gene of Metsulfuron-Methyl Resistant Biotypes Monochoria vaginalis (Burm. f.) C. Presl From West Java, Indonesia," submitted to PLOS ONE under Manuscript ID PONE-D-24-02160. We appreciate the time and effort you have dedicated to reviewing our work.

We are pleased to inform you that we have carefully reviewed your comments and have addressed each point in the attached PDF document. Specifically:

We acknowledge your assessment regarding the significant improvements needed in the Introduction and Discussion sections. As per your comments in the attached PDF labeled PONE-D-24-02160_reviewer, we have extensively revised these sections to improve clarity and coherence.

Based on your suggestion, we have removed the management-related content from the Discussion section to streamline the focus on scientific findings.

Regarding the discussion on NTSR (Non-Target-Site Resistance) and its role in ALS resistance evolution in Monochoria, we have incorporated a detailed analysis in the Discussion section to elucidate these aspects further.

In the Results section, we have provided comprehensive discussions corresponding to the findings presented in Tables and Figures, ensuring clarity and relevance.

Dear Reviewer #2,

Thank you for your insightful feedback.

We have addressed your concern regarding the effects of mutations on the binding and catalytic sites by using the DUET tool to predict protein stability changes. The results of these predictions are included in the paper.

We hope this addition clarifies the impact of the mutations.

We understand the need for significant revisions based on the reviewers' comments. We have undertaken a major revision to address all points raised and ensure our manuscript meets PLOS ONE’s standards.

Thank you again for the opportunity to improve our manuscript. We look forward to resubmitting a revised version that meets PLOS ONE’s publication criteria.

Sincerely,

Kansa Dianti Putri 

IPB University

---

## [Decision Letter · Decision Letter 1]

23 Jul 2024

A New Amino Acid Substitution in the MvALS1 Gene of Metsulfuron-Methyl Resistant Biotypes Monochoria vaginalis (Burm. f.) C. Presl from West Java, Indonesia

PONE-D-24-02160R1

Dear Dr. Kansa Dianti Putri,

We’re pleased to inform you that your manuscript has been judged scientifically suitable for publication and will be formally accepted for publication once it meets all outstanding technical requirements.

Kind regards,

R. M. Sundaram, Ph.D.

Academic Editor

PLOS ONE

Additional Editor Comments (optional):

Based on the comments of the reviewers and based on my own assessment of the revised manuscript, I recommend its acceptance.

Reviewers' comments:

Reviewer's Responses to Questions

**Comments to the Author**

1. If the authors have adequately addressed your comments raised in a previous round of review and you feel that this manuscript is now acceptable for publication, you may indicate that here to bypass the “Comments to the Author” section, enter your conflict of interest statement in the “Confidential to Editor” section, and submit your "Accept" recommendation.

Reviewer #1: All comments have been addressed

Reviewer #2: All comments have been addressed

2. Is the manuscript technically sound, and do the data support the conclusions?

Reviewer #1: Yes

Reviewer #2: Yes

3. Has the statistical analysis been performed appropriately and rigorously? 

Reviewer #1: Yes

Reviewer #2: Yes

4. Have the authors made all data underlying the findings in their manuscript fully available?

Reviewer #1: Yes

Reviewer #2: Yes

5. Is the manuscript presented in an intelligible fashion and written in standard English?

Reviewer #1: Yes

Reviewer #2: Yes

6. Review Comments to the Author

Reviewer #1: (No Response)

Reviewer #2: The authors have revised the manuscript in accordance to my feedbacks. Therefore, I decided to accept this manuscript for potential publication.

7. PLOS authors have the option to publish the peer review history of their article (what does this mean?). If published, this will include your full peer review and any attached files.

Reviewer #1: **Yes: **Simerjeet Kaur

Reviewer #2: No

---

## [Editor Report · Acceptance letter]

30 Jul 2024

PONE-D-24-02160R1 

PLOS ONE

Dear Dr. Putri, 

I'm pleased to inform you that your manuscript has been deemed suitable for publication in PLOS ONE. Congratulations! Your manuscript is now being handed over to our production team.

Kind regards, 

on behalf of

Dr. R. M. Sundaram 

Academic Editor

PLOS ONE